# Technical Note: Tail behaviour of the statistical distribution of extreme storm surges

Tom Howard

Met Office, FitzRoy Road, Exeter, EX1 3PB, UK

**Correspondence:** Tom Howard (tom.howard@metoffice.gov.uk)

**Abstract.**

The tail behaviour of the statistical distribution of extreme storm surges is conveniently described by a return level plot, consisting of water level (Y-axis) against average recurrence interval on a logarithmic scale (X-axis). An average recurrence interval is often referred to as a "return period".

Hunter's allowance for sea-level rise gives a suggested amount by which to raise coastal defences in order to maintain the current level of flood risk, given an uncertain projection of future mean sea-level rise. The allowance is most readily evaluated by assuming that sea-level annual maxima follow a Gumbel distribution, and the evaluation is awkward if we use a generalised extreme value (GEV) fit. When we use a Gumbel fit, we are effectively assuming that the return level plot is a straight line. In other words, the shape parameter, which describes the curvature of the return level plot, is zero.

On the other hand, coastal asset managers may need an estimate of the return period of unprecedented events even under current mean sea levels. For this purpose, curvature of the return level plot is usually accommodated by allowing a non-zero shape parameter whilst extrapolating the return level plot beyond the observations, using some kind of fit to observed extreme values (for example, a GEV fit to annual maxima).

This might seem like a conflict: which approach is "correct"?

Here I present evidence that the shape parameter varies around the coast of the UK, and is consequently not zero.

Despite this, I argue that there is no conflict: a suitably-constrained non-zero-shape fit is appropriate for extrapolation and a Gumbel fit is appropriate for evaluation of Hunter's allowance.

# 1 Introduction

Estimation of the average recurrence interval of unprecedented coastal sea-level events (i.e. events with magnitude even larger than those in the tide gauge record) usually involves a characterisation of the return level plot, which is a plot of water level (Y-axis) against average recurrence interval on a logarithmic scale (X-axis).

This is typically done by fitting a theoretical distribution to observed extreme values. This could mean, for example, fitting a generalised extreme value distribution to the annual maxima, or fitting a Generalised Pareto distribution (GPD) to all peaks over a chosen threshold. The general form of the resulting return level curve is described by the generalised extreme value (GEV) distribution (e.g. Coles, 2001), which has three parameters: location, scale and shape ($\mu, \lambda, \xi$ respectively).

Loosely speaking, the location, scale and shape can be thought of as the intercept, gradient, and curvature of the return level plot (see Howard and Williams, 2021). This technical note is concerned primarily with the *shape parameter*, which, to reiterate, can be thought of as a measure of the *curvature* of the return level plot.

Estimating the average recurrence interval of unprecedented events involves, in effect, extrapolating the return level plot beyond the domain of the observations. For this purpose, there are advantages in terms of simplicity and tractability if we fix the shape parameter at zero, giving the Gumbel distribution and a straight-line return level plot. Coles (2001) and Dixon et al. (1998) counsel against this because the estimated extrapolations, and corresponding uncertainties, are sensitive to the shape parameter (as illustrated by Howard and Williams, 2021, henceforth HW21, their appendix C), and fixing it somewhat arbitrarily at zero can result in substantially underestimated uncertainties. Fixing the shape parameter affects not only the uncertainties, but also the central estimate of the extrapolation (Wahl et al., 2017, see also section 3.1 below).

Martins and Stedinger (2000) note that small-sample maximum-likelihood estimators of the GEV parameters are unstable and recommend use of a Bayesian prior distribution to constrain the shape parameter. From that perspective, the use of the Gumbel distribution could be seen as an overly tight constraint on the shape parameter.

On the other hand, Hunter (2012) proposed a simple allowance for uncertain projected future mean sea-level rise. The method involves treating the expected number of exceedances of a given (high, rare) water level as a cost function. Weighting the sea-level rise projections by this cost function suggests an amount by which to raise coastal defences in order to maintain the current expected costs at some time in the future. Evaluation of Hunter's allowance (Hunter, 2012) in its usual form is simplified by the use of a Gumbel fit, and Woodworth et al. (2021) present evidence in favour of that fit. Furthermore, Van den Brink and Können (2011) also present arguments and evidence in favour of the Gumbel fit, for extreme sea-level data from tide gauges on the Dutch coast.

So, on one hand we have authors arguing in favour of accommodating curvature in the return level plot in order to extrapolate to unprecedented events, and on the other hand we have authors arguing in favour of assuming a straight line in order to simplify the calculation of a sea-level rise allowance. In this note I suggest that these two approaches are compatible.

Skew surge (de Vries et al., 1995) is the difference between the elevation of the predicted astronomical high tide and the nearest (in time) experienced high water. "Experienced high water" here refers to either observed or modelled high water. This short article presents evidence of variations in the GEV shape parameter of skew surge around the coast of the UK.

I note that, in view of this variation, it is generally inappropriate to assume a shape parameter of zero for extrapolation of the return level curve (although, for short record lengths, the shape parameter should be appropriately constrained). However, I show that a Gumbel fit is suitable for the evaluation of Hunter's allowance.

## 2   Introduction to the experiments

This article describes some experiments pertaining to geographical variations in $\xi$ which we found whilst preparing HW21. Two of the experiments were documented in HW21 and are only summarized here. The others are new and are described in detail.

### 2.1   Agreement in $\xi$ from different sources

I believe that the most persuasive evidence we found of geographical variations in $\xi$ is the discovery that the pattern of $\xi$ diagnosed from a simulation correlates strongly with the pattern diagnosed in the most recent UK government guidance, contained in Coastal Flood Boundary Conditions for the UK: update 2018 (Environment Agency, 2018, henceforth CFB2018) from tide gauge data, around the UK coast. Both simulation and tide gauges suggest a large-scale pattern of shape parameter variation including two peaks in the south-west and the south-east with generally slightly lower values on the south coast, and a substantial trough in values around the northern coasts of the mainland (i.e. most of the coast of Scotland). Substantial small-scale variations are superimposed on this large-scale pattern. HW21 (their Fig. 2d and Fig. A1) shows this pattern and the tide gauge locations. Here, a scatter plot (Fig. 1) further illustrates the agreement between these two sources. The shape parameters diagnosed from the simulation (Y-axis) have a negative mean and are generally more negative than those derived from tide-gauge observations (X-axis), which have a positive mean. This is discussed further in HW21.

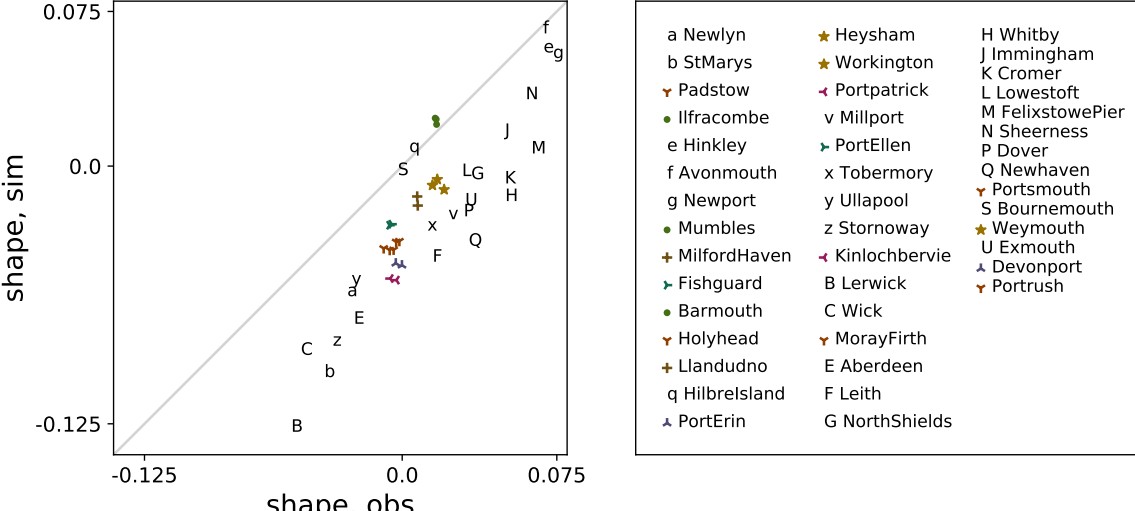

**Figure 1.** Illustrating correlation between shape parameter of skew surge as diagnosed by CFB2018 from tide-gauge data (X-axis) and by HW21 from a simulation (Y-axis), for 44 sites on the UK coast. Pearson's R coefficient is 0.86. Sites which almost overlap on the plot are identified by coloured symbols; other sites are identified by letters. Sites are listed in UK coastal chainage order in the key. Chainage runs clockwise around the UK mainland coast.

The simulation takes atmospheric data from a free-running 484-year climate model control run which does not assimilate any observed data, meaning that the two patterns come from two independent data sources, and yet they correlate well. This suggests that the correlation arises due to factors which are common to both sources: essentially the physics of the atmosphere and ocean (e.g. Pugh and Woodworth, 2014) (for example the surface momentum transfer and the bathymetry of the shelf sea) which are present in reality and represented in a numerical form in the simulation.

However, $\xi$ is not the only parameter to exhibit this spatial correlation between observed and modelled values. Both $\mu$ and $\lambda$ (location and scale parameters) exhibit spatial correlation. When we fit the GEV model to data, we typically find that uncertainties in $\lambda$ and $\xi$ show significant negative correlation: an underestimate in $\lambda$ can be compensated by an overestimate in $\xi$ (Brown, 2021, pers. comm.[1]). This raises the question of whether the spatial correlation between modelled/observed $\xi$ could be an artefact caused by spatial correlation between modelled/observed $\lambda$ somehow "printing through" to create an apparent correlation between modelled/observed $\xi$. To control for this possibility I performed two experiments, which are described in subsection 2.2.

---

[1] This is readily confirmed by generating random Gumbel-distributed samples of record length around, say, 30 (comparable to typical record lengths at UK tide gauge sites) and fitting them by MLE with a GEV fit. In almost all cases the $\lambda, \xi$ covariance will be found to be negative.

## 2.2 Break correlation

To control for the possibility of the $\lambda$—$\xi$ compensation "printing through" into an apparent $\xi$(model, tide-gauge) spatial correlation via the $\lambda$(model, tide-gauge) spatial correlation and the fitting process, I performed two tests intended to break the $\xi$(model, tide-gauge) spatial correlation. In both tests I left the tide-gauge $\xi$ data unchanged. In both cases the results (described below) indicate that the $\xi$(model, tide-gauge) spatial correlation is *not* caused by $\lambda$—$\xi$ compensation.

### 2.2.1 First test

For the model data, at each site, instead of fitting GEV to the annual maxima, I fitted Gumbel to give $(\mu, \lambda)$. Using $(\mu, \lambda)$, I produced random Gumbel "annual maxima" for that site. I then proceeded as before (i.e. fitted a GEV distribution to that random data). If, having done this for all sites, significant $\xi$(random, tide-gauge) spatial correlation is still seen, it could be that the $\xi$(model, tide-gauge) spatial correlation is an artefact of $\lambda$—$\xi$ compensation. No such significant $\xi$(random, tide-gauge) spatial correlation was found.

### 2.2.2 Second test

This test is similar to the first test, but instead of generating random Gumbel data, I generated random GEV data, with spatially shuffled shape parameters. For the model data, at each site, I fitted GEV to the annual maxima to give $(\mu, \lambda, \xi)$. For each site, I used the $(\mu, \lambda)$ of that site, but $\xi$ chosen at random from any site. Using $(\mu, \lambda, \xi)$, I produced random GEV "annual maxima" for that site. I then proceeded as in the first test (i.e. fitted a GEV distribution to that random data). As in the first test, no significant $\xi$(random, tide-gauge) spatial correlation was found.

## 2.3 Woodworth et al. plot applied to model annual maxima

Woodworth et al. (2021) calculate Gumbel scale parameters $\lambda$ for the world coastline, in order to evaluate Hunter's allowance (Hunter, 2012). The Gumbel fit is an obvious choice for this purpose: Hunter's allowance reduces the number of variables in sea-level rise projections, and although the method can be extended to accommodate the GEV, this is at the cost of reintroducing a variable (the allowance then depends on the return level of interest). Howard and Palmer (2020) demonstrate that Hunter's allowance calculations for the UK are not particularly sensitive to curvature of the return level curve, and thus a Gumbel fit is satisfactory for this purpose (see section 3).

Woodworth et al. (2021) use a simple test of the suitability of the Gumbel fit. They use the fact that, for the Gumbel distribution, the scale parameter $\lambda$ is related to the standard deviation by $\lambda = \sigma\sqrt{6}\pi$ where $\sigma$ is the standard deviation of the annual maxima. Thus, they use a plot of $\lambda$ (by maximum-likelihood Gumbel fit to the annual maxima) against $\sigma\sqrt{6}\pi$ (where $\sigma$ is the sample standard deviation) as a goodness-of-fit test.

I followed the approach of Woodworth et al. (2021) using simulated skew surges from the long control run described in HW21. Results are shown in panel (a) of Fig. 2. $\lambda$ estimated from the standard deviation is within the uncertainty of the fitted estimate at most sites, but there is a consistent bias in panel (a) which is absent in the control panel (b), where Gumbel-distributed random data have been used in place of the simulated skew surge data. To quantify this bias, I evaluated the RMS difference across sites between "$\lambda$ from fit" and $\sigma\sqrt{6}\pi$. To quantify the significance of this bias, I made multiple versions of panel (b) (i.e. multiple random samples using a true Gumbel distribution) and evaluated the RMS difference across sites for each version, to give a distribution of RMS difference. The RMS difference based on the real data is more than 6 standard deviations away from the mean of this random distribution, showing the statistical significance of the bias, and re-emphasising that the simulated annual maxima are *not* Gumbel-distributed (for further details see Appendix A).

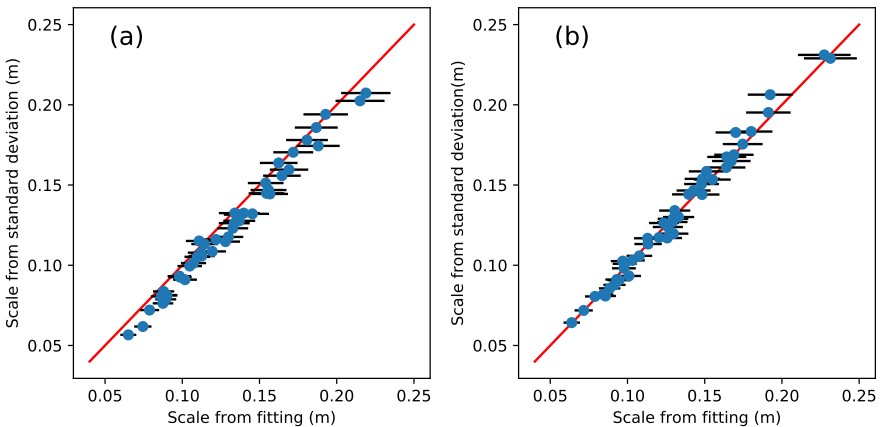

**Figure 2.** (a) Following Woodworth et al. (2021), we compare $\lambda$ diagnosed from a Gumbel fit to 484 years of simulated annual maxima (X-axis) with $\sigma\sqrt{6}/\pi$, where $\sigma$ is the standard deviation of the 484 annual maxima, for UK coastal sites. The two should be the same in the case of the annual maxima conforming to a Gumbel distribution. The horizontal error bars represent 95-percent uncertainty in the fitted $\lambda$. The red line represents a ratio of 1. Each point represents the site of a tide gauge on the coast of the UK. (b) as (a), but instead of simulated annual maxima, at each site we use 484 random variates drawn from the Gumbel distribution, using the $\lambda$ for that site determined by the fit in (a).

## 2.4   Tests following Van den Brink and Können (2008)

Van den Brink and Können (2008) (henceforth VdBK08) devised a sophisticated goodness-of-fit test that emphasises the quality of fit at the most extreme value (the "outlier") from each site. They developed what might be called a "standardised" quantile-quantile (QQ) plot[2] showing the outlier at each site, where "standardised" refers to removing the effects of different

---

[2] A quantile-quantile plot (sometimes "quantile plot" (e.g. Coles, 2001)) typically compares the quantiles (water levels) from two different sources, for example observations on one axis and a statistical model on the other axis.

GEV parameters and record lengths at different sites. The approach is described rigorously in VdBK08. For ease of reference, a brief informal expression of an exactly equivalent procedure follows.

Consider the following concept.

– If the cumulative distribution function (CDF) of a random variate $R$ is known, then a sample $R_i, i = 1...n$ of $n$ values from that distribution can be transformed to a sample $U_i, i = 1...n$ whose expected distribution is standard-uniform[3]. This is sometimes referred to as the probability integral transform (PIT, see for example Folland and Anderson, 2002). If the inverse CDF is known, the PIT can also be used to generate a non-uniformly distributed sample such as $R_i, i = 1...n$ from a standard-uniformly distributed sample such as $U_i, i = 1...n$.

We use this concept in the following procedure.

**Step 1** Suppose that a sample $Y_i, i = 1...n$ of $n$ annual maxima from a given site is assumed to be be drawn from a generalised extreme value (GEV) distribution.

**Step 2** If the parameters of the GEV are known, we can transform this sample using the PIT to a sample $U_i, i = 1...n$ whose expected distribution is standard-uniform. This transformation depends on the parameters.

**Step 3** The "outlier" of this transformed sample, $M = \max(U_i, i = 1...n)$ has its own distribution: $\Pr(M < x) = x^n$ (notice that this depends on $n$).

**Step 4** Since its CDF is known, a sample $M_j, j = 1...m$ of outliers from $m$ different sites can also be transformed to a standard-uniform distribution, $U_j, j = 1...m$, using the PIT. ($n$ need not be the same at every site).

(Reminder: $n$ is the number of annual maxima at a given site. $m$ is the number of sites considered.) We can plot the sorted 145   sample $U_j, j = 1...m$ against its expected (i.e. uniformly-distributed) values. Departures from this expectation will arise due to sampling uncertainty, resulting in departures from the idealised straight line of gradient one and intercept zero.

VdBK08 apply this procedure to a situation where the GEV parameters at each site are *not* known, but rather estimated by fitting a GEV (or Gumbel) to the sample $Y_i, i = 1...n$ at that site. Now departures from the idealised line will arise not only from sampling uncertainty, but also from poor fitting. Thus the departure from the idealised line becomes a measure of goodness-
of-fit, with the emphasis on quality of fit at the outlier at each site. This plot can be thought of as a "standardised" probability-probability (PP) plot[4], where "standardised" refers to removing the effects of different GEV parameters and record lengths at different sites. VdBK08 perform one further transformation. Instead of the plot described above, they take $-\log(-\log(.))$ of both axes. This emphasizes the departures at the high-end extremes of the plot, making their plot more like a "standardised" QQ plot, where again "standardised" refers to removing the effects of different GEV parameters and record lengths at different

---

[3]i.e. uniform between 0 and 1

[4] A probability-probability plot (sometimes "probability plot" (e.g. Coles, 2001)) typically compares the probability of an event as estimated from two different sources, for example estimated empirically from observations on one axis and estimated from a statistical model on the other axis.

sites. It is also known as a Gumbel plot, $-\log(-\log(.))$ being the inverse CDF of the Gumbel distribution. For further details see VdBK08.

VdBK08 show that when unconstrained GEV is fitted to synthetic Gumbel-distributed maxima, the additional free parameter results in over-fitting; the variability in $U_j, j = 1...m$ is seen to be too small.

Among other applications, Van den Brink and Können (2011) used the procedure specifically to help choose between GEV-fitting and Gumbel-fitting the distributions of simulated and observed annual maximum sea level at 19 sites on the Dutch coast. They fixed the shape parameter and found an optimum value of -0.005. They prefer Gumbel-fitting to unconstrained GEV-fitting for this application, arguing that the uncertainties in extrapolation are smaller.

I followed the procedure described to make both types of plot (PP and QQ) for two different sets of annual maximum skew surge at sites of UK tide gauges. The two different sets are tide gauge observations and data from the numerical simulation as used in section 2.1 (for details see Howard and Williams, 2021). The resulting plots are shown in Fig. 3.

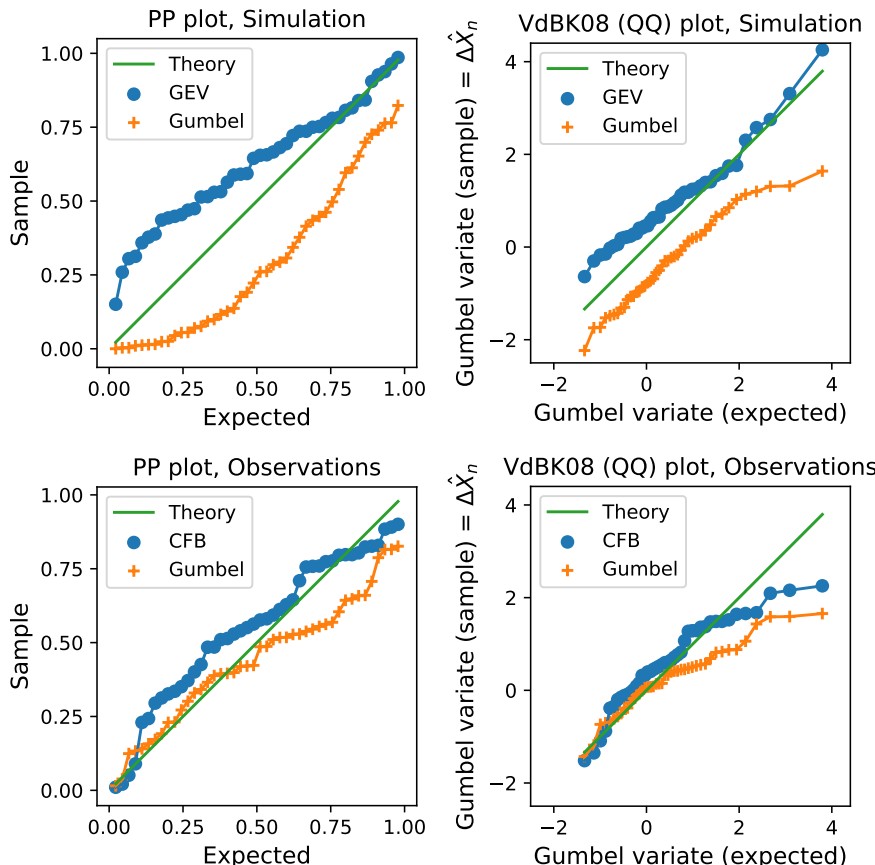

**Figure 3.** Left-hand panels show PP plots as described above and right-hand panels show plots following VdBK08, for skew surge at 44 sites of UK tide gauges (one plotted point per gauge). Orange cross symbols relate to a Gumbel fit to annual maxima. Blue circles relate to unconstrained GEV fits to annual maxima (simulation, top panels) and constrained GPD fits to peaks over a threshold following the CFB2018 guidance (Environment Agency, 2018) method (observations, bottom panels). Each point represents an outlier (the largest annual maximum in the record) at a gauge. The Y-axis of the QQ plot is labelled $\Delta \hat{X}_n$ for consistency with VdBK08.

As in VdBK08 (their figure 3), each plot shows a comparison of two different types of fitting. The top panels compare unconstrained GEV fits (blue circles) to simulated annual maxima with Gumbel fits (orange crosses) to the same annual maxima. These results suggest that allowing for some variation in the shape parameter is preferable to Gumbel fitting, presumably because Gumbel fitting represents too tight a constraint on the shape, for these data.

In the bottom panels, the sophisticated constrained generalised Pareto distribution (GPD) fit to peaks over a threshold (POT) as used in the CFB2018 guidance (Environment Agency, 2018) was applied to observed skew surge extremes, and the blue circles relate to that fit. The orange crosses again relate to simple Gumbel fits to annual maxima. This is a less fair comparison

than that shown in the top panels, because the constrained GPD fit to POT takes advantage of more data. Nevertheless, the constrained GPD fit allows for a non-zero shape parameter whereas the Gumbel fit does not, and again we do not see any clear support for preferring the Gumbel fit.

One possible criticism of this test is that the data are not independent, since a surge event will typically affect several sites. A crude fix is to miss out closely-neighbouring tide gauges, and to use results from every second tide gauge, every third tide gauge, etc. The results of doing so are shown in the Howard and Williams (2021) review material available online (see bibliography for a URL). Again, these results do not provide any clear support for preferring the Gumbel fit.

## 3 Discussion

All of the above experiments illustrate that the shape parameter of sea-level annual maxima around the UK is not, in general, zero, as has been widely recognised (e.g. Marcos and Woodworth, 2017; Wahl et al., 2017; Environment Agency, 2018). Thus it is appropriate to use a (suitably constrained) fit accommodating non-zero shape for the purpose of quantifying the return level curve and extrapolating to quantify levels (and uncertainties) of unprecedented events at these sites.

However, Howard and Palmer (2020) (working with still water level rather than skew surge) have shown that curvature in the return level plot gives variations of less than 5 cm (absolute, and less than 6% relative) in Hunter's allowance for UK coastal sites based on RCP8.5 for 2100, and this contribution to uncertainty in the allowance is less important than the contribution from the uncertainty in the present-day return level curves (see Fig. 4 here, and Howard and Palmer, 2020, their section 5.3). Thus, it is reasonable to use the Gumbel distribution for a simple evaluation of Hunter's allowance, at least for sites on the UK coast. Note, however, that curvature in the return level plot implies that an appropriate allowance may be dependent on the return period of interest (Buchanan et al., 2017).

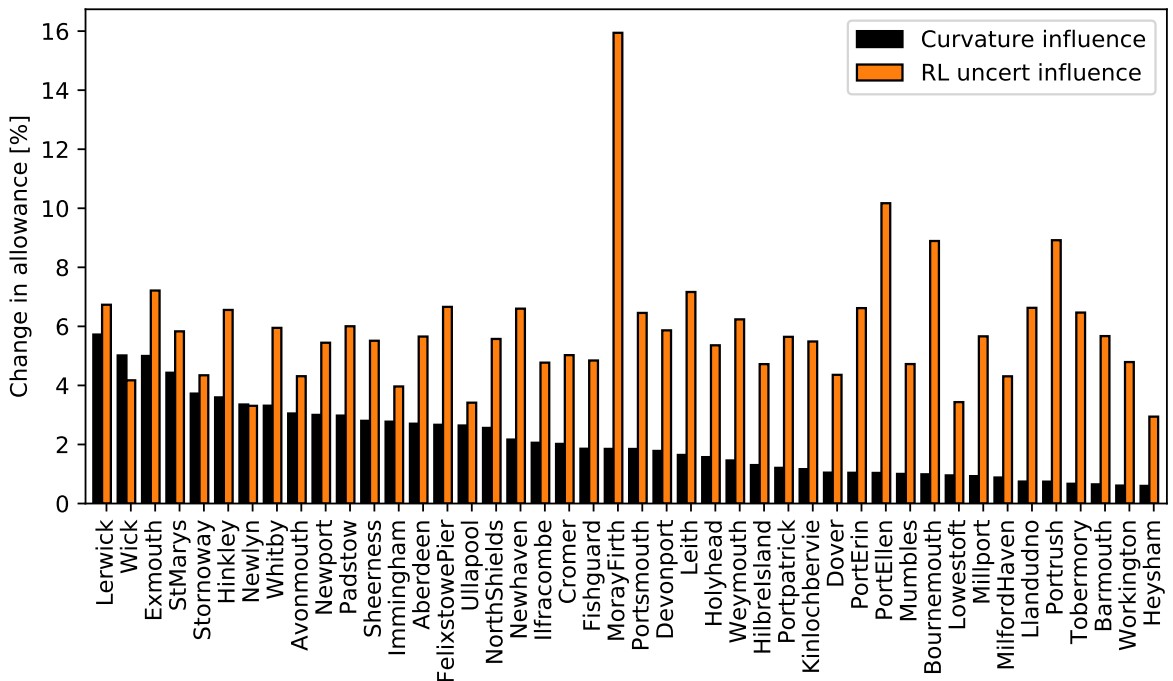

**Figure 4.** Following Howard and Palmer (2020), we show the sensitivity (black filled bars) of Hunter's allowance to curvature of the return level plot of still water level (in other words, sensitivity to the departure of the still water extremes from a Gumbel distribution) for 44 sites on the UK coast. Also shown for comparison is the sensitivity to uncertainty in the return level curve. For full details of the method see Howard and Palmer (2020). Sites are ordered by curvature sensitivity (not geographical location).

## 3.1 Extrapolation

Here are some simplified examples illustrating the effect of constraint when extrapolating the return level curve beyond the observational record. I compare three different parametric fits to observed annual maximum skew surge at UK coastal sites:

1. GEV fit unconstrained

     2. GEV fit constrained to give a range of shape parameters comparable to CFB2018.

     3. Gumbel fit

Maximum likelihood estimation (MLE) is used in all cases (Generalised MLE (Martins and Stedinger, 2000) for case 2). CFB2018 used a normal prior on the shape with mean 0.0119 and standard deviation 0.0343. Applying this to only the annual 200    maxima gives a much smaller range of shapes than that found by CFB2018. This is because the annual maxima represent a

smaller data set than the peaks over threshold used by CFB2018, and this smaller quantity of data allows the prior to dominate. To compensate for this, I increased the width of the prior until I produced a similar range of shapes to CFB2018. My prior has mean 0.0119 and standard deviation 0.062. Using only the annual maxima is not to be recommended in general, because better results can be obtained using more data (for example N-largest or peaks-over-threshold). Nevertheless, I take the constrained

GEV fit to be "truth" for the purpose of this illustration, and identify anomalies relative to that. Anomalies are shown in fig. 5.

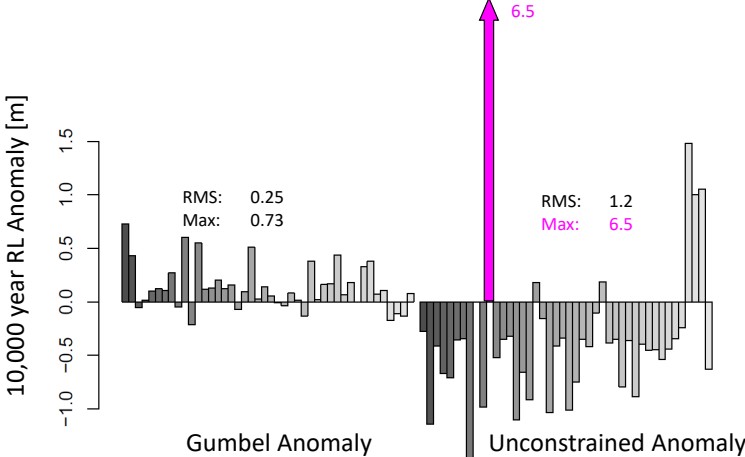

**Figure 5.** Anomalies in the ten thousand year return level of skew surge as estimated from annual maxima at 44 UK coastal sites by maximum likelihood estimation using a Gumbel fit (left) and an *unconstrained* GEV fit (right). Anomalies are relative to a *constrained* GEV estimate (see main text). The bar shown in pink (Hinkley Point) exceeds the limits of the plot.

This figure shows that, even though we believe that the data represent distributions with non-zero shape parameters, the likely inaccuracies associated with unconstrained shape parameters are more serious than the likely inaccuracies associated with the over-constraint of insisting the shape parameters be zero (Gumbel fitting). In other words, we see the importance of choosing an appropriate prior constraint on the shape parameter, for typical real-world record lengths.

The most serious anomaly in the unconstrained GEV fit is at Hinkley Point in the Bristol Channel, where only 26 skew surge annual maxima from the tide-gauge record are included in the fit. The difficulty with this site was noted and discussed by Batstone et al. (2013). Figure 6 illustrates the three different fitted return level curves at that site. Also shown is an example site (Bournemouth, 18 annual maxima), where the unconstrained shape parameter is negative.

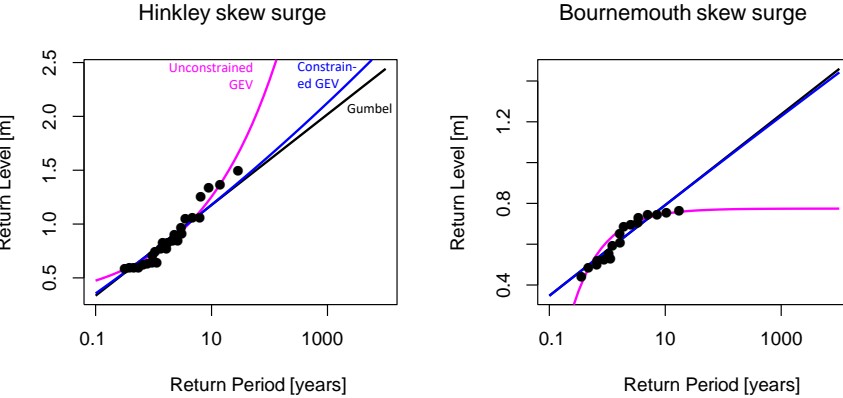

**Figure 6.** Illustrating a well-known issue with unconstrained GEV fit by MLE to annual maxima from a short record. The three different fits are described in the main text.

We have seen in previous sections that the shape parameter of UK skew surges varies spatially and, consistent with this, we
have some confirmation (in section 2.4) that an appropriately constrained GEV fit is preferable to a Gumbel fit for UK skew surges. Figures 3, 5 and 6 all show that the Gumbel fit is at odds with an appropriately constrained GEV fit, but on the other hand, Figs. 5 and 6 show that the Gumbel fit is much safer than an unconstrained GEV fit to short records.

## 4   Conclusions

In summary, at least for sites on the UK coast, a non-zero shape parameter should be accommodated at the fitting stage for the
purpose of extrapolating the storm surge return level curve. However, fitting with an unconstrained shape parameter to short records is not advisable, as it is liable to give larger errors than the over-constraint inherent in a Gumbel fit.

Also, it is reasonable to use a Gumbel fit for evaluation of Hunter's allowance.

*Data availability.* Contains Environment Agency information ©Environment Agency and database right. The Environment Agency CFB2018 technical report is available to download from https://environment.data.gov.uk
All data used in the figures here are available in the supplementary material.

## Appendix A: Statistical test associated with figure 2

Let $m$ be the number of sites considered.

Let $\{Y_1, Y_2, ... Y_i, ... Y_n\}$ be the annual maximum skew surges over $n$ years at a given site. In the context of figure 2 we are considering annual maxima simulated by our numerical model of the shelf sea, and $n = 484$.

Let $\lambda_j$ be the scale parameter diagnosed by a Gumbel fit to the $n$ annual maxima at site $j$, let $\sigma_j$ be the standard deviation of the $n$ annual maxima at site $j$, and let

$$d_j = \lambda_j - \sigma_j \frac{\sqrt{6}}{\pi}$$

This is the departure of a point in figure 2 panel (a) from the line $x = y$. Then our test metric (call it $T_Y$) is the root-mean-square value of $\{d_1, d_2, ... d_j, ... d_m\}$:

$$T_Y = \sqrt{\overline{d_j^2}}$$

where the overbar indicates a mean over $j = 1, 2, ... m$ sites. This test metric is a single value. To test the statistical significance of the value of $T_Y$ that we find when $\{Y_1, Y_2, ... Y_i, ... Y_n\}$ are the annual maximum skew surges simulated by our numerical model of the shelf sea, we repeat the test, replacing $\{Y_1, Y_2, ... Y_i, ... Y_n\}$ at each site $j$ by $\{G_1, G_2, ... G_i, ... G_n\}$ where $\{G_1, G_2, ... G_i, ... G_n\}$ is a random sample drawn from a Gumbel distribution whose scale parameter[5] is $\lambda_j$. We do this many times (say $N = 256$ times) to create a 256-element distribution of values $\{T_{G,1}, T_{G,2}, ... T_{G,k}, ... T_{G,N}\}$, each being a value of $T_G$ that we find when $\{G_1, G_2, ... G_i, ... G_n\}$ are "easily-made pseudo-extremes" from a Gumbel distribution, instead of 'hard-won" simulated annual maxima of unknown distribution like $\{Y_1, Y_2, ... Y_i, ... Y_n\}$. The variation in $\{T_{G,1}, T_{G,2}, ... T_{G,k}, ... T_{G,N}\}$ arises due to sampling uncertainties.

When we compare $T_Y$ with the distribution $\{T_{G,1}, T_{G,2}, ... T_{G,k}, ... T_{G,N}\}$, we find that $T_Y$ departs from the mean of $\{T_{G,1}, T_{G,2}, ... T_{G,k}, ... T_{G,N}\}$ by more than six times the standard deviation of $\{T_{G,1}, T_{G,2}, ... T_{G,k}, ... T_{G,N}\}$, implying that this departure is not simply an artefact of sampling, but arises from the fact that the $\{Y_1, Y_2, ... Y_i, ... Y_n\}$ are *not* Gumbel-distributed. This large departure (more than six standard deviations) might seem surprising given that figure 2 (a) shows that most points are within the 95-percent (approximately 2 standard deviation) uncertainty range of the $x = y$ line. The large departure is associated with the fact that the shape parameters of the $\{Y_1, Y_2, ... Y_i, ... Y_n\}$ are predominantly negative. We can see this expressed in figure 2 panel (a), where almost all of the scatter points lie to the right of the line $x = y$, whereas in panel (b) (which is one of the 256 examples of what happens when we replace the $Y_i$ by $G_i$) the scatter points lie either side of the line.

It would be interesting to apply a similar statistical test to the scatter of points in figure S1 of the supplementary material to Woodworth et al. (2021). Assuming that, as found by Wahl et al. (2017), the shape parameters are predominantly negative,

---

[5] We could also employ the corresponding location parameter, but there is no need because this only introduces an offset. We simply set the location parameter to zero. Incidentally, we can generate this random sample from a uniformly-distributed random sample using the probability integral transform, among other possible approaches.

one might expect to see the test statistic $T_Y$ similarly outside the distribution $T_{G,k}, k = 1, 2, ...N$, although the shortness of the

tide-gauge records might reduce the statistical significance. The bias, $\overline{d_j}$, is another alternative test statistic.

*Author contributions.* TH performed the tests and wrote the text. Please also see the acknowledgements.

*Competing interests.* The author declares no competing interests.

*Acknowledgements.* This work was supported by the Met Office Hadley Centre Climate Programme funded by the UK Government's Department for Business, Energy and Industrial Strategy and Department for Environment, Food and Rural Affairs. Thanks to Phil Woodworth,

John Hunter, and an anonymous referee, who all contributed helpful review comments. Thanks to Simon Williams and Jenny Sansom for help with the CFB data.

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
