# Peer review of "Technical Note: Tail behaviour of the statistical distribution of"

_Ocean Science, 2022_

## Author Comment (AC1)

|   | Comments on "Technical Note: Tail      |                                                       |
|---|----------------------------------------|-------------------------------------------------------|
|   | behaviour of the statistical           |                                                       |
|   | distribution of extreme storm          |                                                       |
|   | surges" by Tom Howard (OSD)            |                                                       |
|   | This is a short technical note which   | Thank you for your thorough review. I have responded  |
|   | attempts to make 3 points: (a) the     | to each point as tabulated below, in most cases       |
|   | shape parameter of extreme sea         | modifying the manuscript.                             |
|   | level curves at most UK sites is not   |                                                       |
|   | zero (and usually negative) and so     |                                                       |
|   | any parameterisation of the            |                                                       |
|   | extreme level curve should             |                                                       |
|   | accommodate its curvature, (b) in      |                                                       |
|   | spite of that, an assumption of zero   |                                                       |
|   | shape for a Gumbel distribution is     |                                                       |
|   | reasonable for Hunter's allowance      |                                                       |
|   | calculation, and (c) the shape         |                                                       |
|   | parameters derived from short          |                                                       |
|   | records are imprecise. These things    |                                                       |
|   | were known already (or suspected       |                                                       |
|   | anyway) but it does no harm to         |                                                       |
|   | restate them in the same place.        |                                                       |
|   | I have no objections to the note's     |                                                       |
|   | publication if the small things        |                                                       |
|   | below can be attended to. The text     |                                                       |
|   | is clearly written although the        |                                                       |
|   | document itself is a little rough      |                                                       |
|   | (hence some of the trivial             |                                                       |
|   | comments below).                       |                                                       |
| 1 | line 6 - mean sea level rise here and  | I have attempted to improve the consistency,          |
|   | mean-sea-level rise at line 40 (I said | hyphenating "sea level" only when it is a compound    |
|   | these were trivial comments but        | adjective.                                            |
|   | they suggest some lack of              |                                                       |
|   | attention)                             |                                                       |
| 2 | 15 - you don't present evidence        | Have included a text description of the large-scale   |
|   | that the shape parameter varies        | pattern and a more clear indication of where the      |
|   | around the UK coast. You have a        | pattern (and a map) are shown in HW21.                |
|   | scatter plot in Figure 1 that shows    |                                                       |
|   | there are clearly different values at  |                                                       |
|   | different places but, unless you       |                                                       |
|   | know where the UK place names          |                                                       |
|   | refer to, you have no insight on       |                                                       |
|   | how the shape varies around the        |                                                       |
|   | actual coastline. A map is needed      |                                                       |
|   | or at least a couple of sentences to   |                                                       |
|   | say how it varies.                     |                                                       |
| 3 | 49 - not incompatible ==>              | Modified.                                             |
|   | compatible!                            |                                                       |
| 4 | 50-52 - these lines would be better    | Yes - done.                                           |
|   | following on at line 39                |                                                       |
| 5 | 65 - I know this is a short technical  | The surge extrema are not "from the same years" at    |
|   | note and there are many details in     | all. The simulation is a "free-running" climate model |

|   | HW21, but it does no harm to give
some essential minimum
information. For example,
presumably the surge extrema used
for Figure 1 are from exactly the
same years as the tide gauge
extrema, or comparisons are not
exact. So say so. Also say what the
minimum record length of tide
gauge record is employed.                                                                                                                                                                                                                                                                       | control run - all of the atmospheric data is simulated.
That is why I find it so interesting/surprising that the
shape parameters correlate well with those diagnosed
from the tide gauges. I have modified the text to make
this clearer. I'm not sure that the length of the tide
gauge records used is relevant to this particular
result? I think it is more relevant to Fig.5, where it is
mentioned.                                                                                                                                                                                                                                                                                                                                                                                                                                                                                                                                                                                                                                                                                                                                                                                          |
|---|------------------------------------------------------------------------------------------------------------------------------------------------------------------------------------------------------------------------------------------------------------------------------------------------------------------------------------------------------------------------------------------------------------------------------------------------------------------------------------------------------------------------------------------------------------------------------------------------------------------|--------------------------------------------------------------------------------------------------------------------------------------------------------------------------------------------------------------------------------------------------------------------------------------------------------------------------------------------------------------------------------------------------------------------------------------------------------------------------------------------------------------------------------------------------------------------------------------------------------------------------------------------------------------------------------------------------------------------------------------------------------------------------------------------------------------------------------------------------------------------------------------------------------------------------------------------------------------------------------------------------------------------------------------------------------------------------------------------------------------------------------------------------------------------------------------------------------------------------|
| 6 | After Figure 1 there should be a
sentence to tell the reader that
most of the UK shape parameters
are negative. And that this
observation is
not new. For example, see Figure 9
of Marcos and Woodworth (JGR,
2017) which shows consistent
negative shape parameters for both
North Atlantic coasts. And Wahl et
al. (2017) claim that 85% of records
worldwide have negative shape
parameters. As for the UK, I am
sure the negative shapes will have
been pointed out in older papers by
Blackman, Horsburgh, Tawn etc.
(although I have not checked
which) | Thank you for reminding me of these two very
relevant papers, both of which are cited elsewhere in
the manuscript. But, regarding the shape parameters
at UK tide gauges as discussed here, are they known
to be negative? The recent CFB2018 report (with
project team including Jon Tawn and Kevin
Horsburgh) diagnosed a mixture of positive and
negative shape parameters. I am not aware of a more
recent publication specific to the UK which overturns
that result.                                                                                                                                                                                                                                                                                                                                                                                                                                                                                                                                                                                                                                                                                                                    |
| 7 | 69 - give a reference. For example
chapter 7 of Pugh and Woodworth
(2014). As well as the physics of
wind stress etc., there is a general
point that there is only so much
water in the ocean, so one would
imagine any extreme level curve to
turn down at some point.                                                                                                                                                                                                                                                                                                                     | Reference: done. Regarding the curve turning down: I
agree. A similar argument would apply to any physical
process, I think. I contacted Simon Brown (a Met
Office climate extremes expert) about this. He said:
"Ultimately all meteorological processes have a
physical limit and so should have bounded
distributions. However, we don't have perfect
samples and we don't fit perfect statistical models.
Both, I would argue, can lead to unbounded models
seeming to be the best fit to data. As you say there
seems to be no consensus on what to do about this
nor even much of a discussion about the problem.
My go to example is with wind extremes. An
observational record will consist of samples of the
warm conveyor belt, the cold conveyor belt and if you
are lucky a sting jet. Each of these processes have
different extreme behaviour but we fit a simple EV
distribution. If there is just one sting jet sample,
which is way above the others, the resultant EV fit will
have a positive shape parameter even if all the three
sub-processes have negative shape parameters. The
fitted EV model does not reflect the underlying |

|    |                                                                                                                                                                                 | mixture of processes and so when it is fitted we get a                                                                                                                                                                                                                                                                                                                                                                                                 |
|----|---------------------------------------------------------------------------------------------------------------------------------------------------------------------------------|--------------------------------------------------------------------------------------------------------------------------------------------------------------------------------------------------------------------------------------------------------------------------------------------------------------------------------------------------------------------------------------------------------------------------------------------------------|
|    |                                                                                                                                                                                 | distribution that looks unphysical.                                                                                                                                                                                                                                                                                                                                                                                                                    |
|    |                                                                                                                                                                                 | What happens is that generally there is some
pragmatic choice that seems to fit the main objective
on the analysis. I suppose being a good scientist one
would look at the results with a free shape fit and
compare with a constrained shape fit and discuss the
merits of each. This can be quite subtle – with my
wind example it is not clear that forcing the shape
parameter to be <=0 will give you a more physical fit if |
|    |                                                                                                                                                                                 | the error of not including the mixture aspect of the                                                                                                                                                                                                                                                                                                                                                                                                   |
| 8  | 73 in [shape parameter]                                                                                                                                                         | L have added a "pers, comm." type citation and a                                                                                                                                                                                                                                                                                                                                                                                                       |
|    | (reference needed. HW21 again?)                                                                                                                                                 | footnote.                                                                                                                                                                                                                                                                                                                                                                                                                                              |
|    |                                                                                                                                                                                 | A colleague (Simon Brown again) pointed me to the scale-shape compensation issue last year. He can no                                                                                                                                                                                                                                                                                                                                                  |
|    |                                                                                                                                                                                 | read it. I had thought to maybe show analytically that
the partial cross-derivative of the loglikelihood is
usually negative, but it proved to be beyond my
mathematical scope. So, I have settled for a footnote
explaining that it can readily be confirmed
numerically, and how to do so.                                                                                                                                            |
| 9  | I don't understand why in practice                                                                                                                                              | Yes, I do see what you mean. I have modified the                                                                                                                                                                                                                                                                                                                                                                                                       |
|    | you know there is spatial                                                                                                                                                       | caption of Fig 1 to clarify that we are dealing with the                                                                                                                                                                                                                                                                                                                                                                                               |
|    | parameters. That can only be in                                                                                                                                                 | dependent on the datum.                                                                                                                                                                                                                                                                                                                                                                                                                                |
|    | model runs where the datum at                                                                                                                                                   |                                                                                                                                                                                                                                                                                                                                                                                                                                                        |
|    | every point is MSL. But if you are                                                                                                                                              |                                                                                                                                                                                                                                                                                                                                                                                                                                                        |
|    | using real tide gauge data the                                                                                                                                                  |                                                                                                                                                                                                                                                                                                                                                                                                                                                        |
|    | the datums used at each site (I                                                                                                                                                 |                                                                                                                                                                                                                                                                                                                                                                                                                                                        |
|    | hope you see what I mean.)                                                                                                                                                      |                                                                                                                                                                                                                                                                                                                                                                                                                                                        |
| 10 | 79 - why 'vector'? It seems an odd word to use here.                                                                                                                            | Now "data"                                                                                                                                                                                                                                                                                                                                                                                                                                             |
| 11 | 82 - say 'For the model data at each
tide gauge site'. To make it clear
you are using just the short model
data sets here and not the 484 year
set mentioned later. | I am using the 484 year data, There is no short model
data set. Have modified the m/s: "The simulation
takes atmospheric data from a free-running 484-year
climate model control run"                                                                                                                                                                                                                                                         |
| 12 | 90 - from any other site. (?)                                                                                                                                                   | Actually from any site. Could in theory be the same                                                                                                                                                                                                                                                                                                                                                                                                    |
|    |                                                                                                                                                                                 | one very occasionally, but since they are shuffled at
random many times this does not materially affect the
result.                                                                                                                                                                                                                                                                                                                              |
| 13 | 104 - the long run of 484 years. And this is for the 44 (?) tide gauge sites                                                                                                    | Yes. Have adjusted caption.                                                                                                                                                                                                                                                                                                                                                                                                                            |
| 14 | 112 not Gumbel-distributed as was known previously.                                                                                                                             | Changed to "re-emphasising"                                                                                                                                                                                                                                                                                                                                                                                                                            |
| 15 | Figure 2 (a) and (b) should have (m) on each axis                                                                                                                               | Thank you.                                                                                                                                                                                                                                                                                                                                                                                                                                             |

| 16 | line 4 of caption the site of the                                                                                                                                                                                                                                                                                                                                                                            | Yes. Have adjusted caption.                                                                                                                                                                                                                                                                                                                                                                                                                                                                                                                                                                                                                                                                     |
|----|--------------------------------------------------------------------------------------------------------------------------------------------------------------------------------------------------------------------------------------------------------------------------------------------------------------------------------------------------------------------------------------------------------------|-------------------------------------------------------------------------------------------------------------------------------------------------------------------------------------------------------------------------------------------------------------------------------------------------------------------------------------------------------------------------------------------------------------------------------------------------------------------------------------------------------------------------------------------------------------------------------------------------------------------------------------------------------------------------------------------------|
| 47 | 44 (?) tide gauges on                                                                                                                                                                                                                                                                                                                                                                                        |                                                                                                                                                                                                                                                                                                                                                                                                                                                                                                                                                                                                                                                                                                 |
| 17 | section 2.4 - I got the idea of this
section although you have to read it
a few times. It would help to fully
explain things. For example, what
does 'standard-uniform' (line 121)
mean?                                                                                                                                                                                                      | Have added a note explaining "standard uniform", and
reworded parts of the section. Have also tried to
further emphasise that this is just a short informal
description, for ease of reference, of the procedure
that VdBK08 used. Have directed the reader to
VdBK08 for full details.                                                                                                                                                                                                                                                                                                                                                                                          |
| 18 | 126 from a given site conforms
to a precise GEV distribution.                                                                                                                                                                                                                                                                                                                                             | Reworded.                                                                                                                                                                                                                                                                                                                                                                                                                                                                                                                                                                                                                                                                                       |
| 19 | 130 depends on the three GEV parameters.                                                                                                                                                                                                                                                                                                                                                                     | Reworded.                                                                                                                                                                                                                                                                                                                                                                                                                                                                                                                                                                                                                                                                                       |
| 20 | 133-134 - standard-uniform (as
above)                                                                                                                                                                                                                                                                                                                                                                     | Thank you.                                                                                                                                                                                                                                                                                                                                                                                                                                                                                                                                                                                                                                                                                      |
| 21 | 151 - an average (?) optimum
They preferred                                                                                                                                                                                                                                                                                                                                                               | They use the word "optimal". I believe they
performed their test with a fixed shape parameter
across all sites, varying the shape parameter to see
which value gave the "best" plot (closest, in some
sense, to x=y on their plot).                                                                                                                                                                                                                                                                                                                                                                                                                                                 |
| 22 | 155 - simulation as represented in
Figure 1 (presumably)                                                                                                                                                                                                                                                                                                                                                  | Yes. Reworded.                                                                                                                                                                                                                                                                                                                                                                                                                                                                                                                                                                                                                                                                                  |
| 23 | Figure 3 - I don't understand why
there are 4 plots here. Shouldn't
there be 8? You have tide gauge
data (shown here) and line 155 says
you use model data also, so you
need another 4 for the model data?                                                                                                                                                                                    | Figure 3 shows simulation-based results in top two
panels and observation-based results (i.e. tide-gauge-
based) in bottom two panels.                                                                                                                                                                                                                                                                                                                                                                                                                                                                                                                                                    |
| 24 | title caption should be VdBK and
not VdB&K to be consistent with
the text But I would remove that
anyway and just have QQ plot to be
consistent with PP plot on the right.
Preumably the dots are ordered so
as to be monotonic. Define in the
caption the delta symbol on y-axis
for QQ (differences at the outliers).
Finally I don't understand why you
call them 'theory'. | Have changed to VdBK08 to be consistent with the text. Yes, the dots are ordered so as to be monotonic (i.e. they are ranked). This is the usual approach in this kind of plot, as I understand it. The X-values of the PP plot are the expected values for a ranked set (of size m) of standard-uniform data:
i/(m+1) (i: 1 to m). The Y-values are what we have for our sample set, given the fit. The QQ plot in this case is the same data, with both X- and Y-values transformed by -log(-log(.)) as described in the text. I have used the word "theory" for consistency with VdBK08: it shows the relationship you would see if every element of your sample took its expected value. |
| 25 | caption line 1 - this should be
reworded as you say above for both
that QQ and PP derive from VdBK                                                                                                                                                                                                                                                                                                     | They are not both derived from VdBK08. See the line
in the text which says "Instead of the plot described
above, they"
VdBK08 show only what I call the QQ plot. They do
not show the PP plot.                                                                                                                                                                                                                                                                                                                                                                                                                                                                                      |
| 26 | caption line 2 - at the 44 (?) sites of UK                                                                                                                                                                                                                                                                                                                                                                   | Yes. Done.                                                                                                                                                                                                                                                                                                                                                                                                                                                                                                                                                                                                                                                                                      |
| 27 | 172 - in general zero, as known
already (refs).                                                                                                                                                                                                                                                                                                                                                           | Added.                                                                                                                                                                                                                                                                                                                                                                                                                                                                                                                                                                                                                                                                                          |
| 28 | Figure 4 - I thought you were using 44 sites (see caption figure 5). This                                                                                                                                                                                                                                                                                                                                    | Amended: now 44.                                                                                                                                                                                                                                                                                                                                                                                                                                                                                                                                                                                                                                                                                |

|   |    | should be mentioned at the places                                                                                                                                   |                                                                                                                                                                                                                                                                                                                                                                                                                                                                                                                                                                                                                                                                                                                                                                     |
|---|----|---------------------------------------------------------------------------------------------------------------------------------------------------------------------|---------------------------------------------------------------------------------------------------------------------------------------------------------------------------------------------------------------------------------------------------------------------------------------------------------------------------------------------------------------------------------------------------------------------------------------------------------------------------------------------------------------------------------------------------------------------------------------------------------------------------------------------------------------------------------------------------------------------------------------------------------------------|
|   |    | in the text I pointed out above.                                                                                                                                    |                                                                                                                                                                                                                                                                                                                                                                                                                                                                                                                                                                                                                                                                                                                                                                     |
|   |    | However here in Figure 4 there are                                                                                                                                  |                                                                                                                                                                                                                                                                                                                                                                                                                                                                                                                                                                                                                                                                                                                                                                     |
| - |    | 46 locations given.                                                                                                                                                 |                                                                                                                                                                                                                                                                                                                                                                                                                                                                                                                                                                                                                                                                                                                                                                     |
|   | 29 | 188 - why did CFB2018 take
+0.0119 as its prior shape
parameter when all the evidence
from previous publications and
your Figure 1 has it negative? And | Good question! This is discussed by Jon Tawn and
Eleanor D'Arcy in section 3.3 ("Penalised Likelihood ")
of their comments on HW21, which are in the public
domain at                                                                                                                                                                                                                                                                                                                                                                                                                                                                                                                                                                                      |
|   |    | of +0.0119 ?                                                                                                                                                        | https://nhess.copernicus.org/preprints/nhess-2021-
184/nhess-2021-184-RC3-supplement.pdf                                                                                                                                                                                                                                                                                                                                                                                                                                                                                                                                                                                                                                                                         |
|   |    |                                                                                                                                                                     | The relevant paragraph says:
"The prior that was selected in the CFB2018 work was
not subjective in the traditional sense of a subjective
prior in Bayesian methods. It was actually a data-
based prior which corresponds to an empirical
Bayesian prior, using all the information that
separately estimated shape parameters for UK skew
surge provide. The effect of this was simply to move
shape parameter estimates more towards the UK
average, with the larger changes coming for sites
with shorter record lengths."                                                                                                                                                                                                        |
|   |    |                                                                                                                                                                     | Considering the scatter plot (Figure 1 of the
manuscript under review here), as you say an overall
negativity is seen in the shape parameters. But that is
only in the shape parameters diagnosed from the
simulation (Y-axis), and not from the tide gauges (X-
axis). Even without the penalty function, the
unconstrained shape parameter estimates based on
the tide gauge data have a positive mean (+0.0119).
Their estimates are shown in figure E.1 of the CFB
2018 report
https://www.gov.uk/government/publications/coastal
-flood-boundary-conditions-for-uk-mainland-and-
islands-design-sea-levels
Their prior is formed from the distribution of their
unconstrained estimates. |
|   |    |                                                                                                                                                                     | I used a prior with a mean of +0.0119 to be
consistent with the CFB2018 approach.
Incidentally, the negative bias in the simulation-based
shape parameters compared to the observation-
based remains unexplained. It is discussed at some
length in HW21. I have checked that it is not simply a
sign issue (unfortunately authors and statistics
packages use differing conventions regarding what is
meant by a positive shape parameter).                                                                                                                                                                                                                                                                                               |
|   |    |                                                                                                                                                                     |                                                                                                                                                                                                                                                                                                                                                                                                                                                                                                                                                                                                                                                                                                                                                                     |

| 30 | Could you explain Figure 5 a bit
better? If the data really has a non-
zero shape, and the choice of prior
is reasonable, then wouldn't you
expect the right-hand side to be
tighter than the left for Gumbels? | I have added the following phrase: "This figure shows
that, even though we believe that the data represent
distributions with non-zero shape parameters, the
likely inaccuracies associated with unconstrained
shape parameters are more serious than the likely
inaccuracies associated with the over-constraint of
insisting the shape parameters be zero (Gumbel
fitting). In other words, we see the importance of
choosing an appropriate prior constraint on the shape
parameter, for typical real-world record lengths."
As the existing text already explains, we are choosing
to take the constrained fit as "truth" for the purpose
of this experiment. |
|----|--------------------------------------------------------------------------------------------------------------------------------------------------------------------------------------------------------------------------------|-------------------------------------------------------------------------------------------------------------------------------------------------------------------------------------------------------------------------------------------------------------------------------------------------------------------------------------------------------------------------------------------------------------------------------------------------------------------------------------------------------------------------------------------------------------------------------------------------------------------------------------------------------------------------------------------------------|
| 31 | 198 is negative in common with
most UK sites (Figure 1) and
worldwide (Marcos and
Woodworth, 2017; Wahl et al.,
2017).                                                                                             | Again, I'm not so confident about asserting that here.
As mentioned above in response to your comment
number 29, the negative shape parameters in my
Figure 1 are only for the simulation. See also Fig E.1 of
the CFB2018 publication: they diagnose a fairly
balanced distribution of +ve and -ve                                                                                                                                                                                                                                                                                                                                                                                    |
| 32 | Figure 6 left - the Hinkley plot is
described in some detail in
Batstone et al. (2013)                                                                                                                                   | Thanks. I have added a citation. Although,
incidentally, I regard Batstone et al. as having been
largely superseded by CFB2018.                                                                                                                                                                                                                                                                                                                                                                                                                                                                                                                                                                 |
| 33 | Acknowledgements - define BEIS
and Defra                                                                                                                                                                                    | Done.                                                                                                                                                                                                                                                                                                                                                                                                                                                                                                                                                                                                                                                                                                 |
| 34 | 221 - Climate, 231 - Research
Letters, 235 - Communications                                                                                                                                                                 | Amended. Thanks.                                                                                                                                                                                                                                                                                                                                                                                                                                                                                                                                                                                                                                                                                      |

---

## Author Comment (AC3)

Author responses to three reviews of "Technical Note: Tail behaviour of the statistical distribution of extreme storm surges" by Tom Howard (OSD).

| Ref. | Reviewer 1: Phil Woodworth | |
|------|---------------------------|---|
| | This is a short technical note which attempts to make 3 points: (a) the shape parameter of extreme sea level curves at most UK sites is not zero (and usually negative) and so any parameterisation of the extreme level curve should accommodate its curvature, (b) in spite of that, an assumption of zero shape for a Gumbel distribution is reasonable for Hunter's allowance calculation, and (c) the shape parameters derived from short records are imprecise. These things were known already (or suspected anyway) but it does no harm to restate them in the same place. | Thank you for your thorough review. I have responded to each point as tabulated below, in most cases modifying the manuscript. |
| | I have no objections to the note's publication if the small things below can be attended to. The text is clearly written although the document itself is a little rough (hence some of the trivial comments below). | |
| 1.1 | line 6 - mean sea level rise here and mean-sea-level rise at line 40 (I said these were trivial comments but they suggest some lack of attention) | I have attempted to improve the consistency, hyphenating "sea level" only when it is a compound adjective. |
| 1.2 | 15 - you don't present evidence that the shape parameter varies around the UK coast. You have a scatter plot in Figure 1 that shows there are clearly different values at different places but, unless you know where the UK place names refer to, you have no insight on how the shape varies around the actual coastline. A map is needed or at least a couple of sentences to say how it varies. | Have included a text description of the large-scale pattern and a more clear indication of where the pattern (and a map) are shown in HW21. |
| 1.3 | 49 - not incompatible ==> compatible! | Modified. |
| 1.4 | 50-52 - these lines would be better following on at line 39 | Yes - done. |

| 1.5 | 65 - I know this is a short technical note and there are many details in HW21, but it does no harm to give some essential minimum information. For example, presumably the surge extrema used for Figure 1 are from exactly the same years as the tide gauge extrema, or comparisons are not exact. So say so. Also say what the minimum record length of tide gauge record is employed. | The surge extrema are not "from the same years" at all. The simulation is a "free-running" climate model control run - all of the atmospheric data is simulated. That is why I find it so interesting/surprising that the shape parameters correlate well with those diagnosed from the tide gauges. I have modified the text to make this clearer. I'm not sure that the length of the tide gauge records used is relevant to this particular result? I think it is more relevant to Fig.5, where it is mentioned. |
|---|---|---|
| 1.6 | After Figure 1 there should be a sentence to tell the reader that most of the UK shape parameters are negative. And that this observation is not new. For example, see Figure 9 of Marcos and Woodworth (JGR, 2017) which shows consistent negative shape parameters for both North Atlantic coasts. And Wahl et al. (2017) claim that 85% of records worldwide have negative shape parameters. As for the UK, I am sure the negative shapes will have been pointed out in older papers by Blackman, Horsburgh, Tawn etc. (although I have not checked which) | Thank you for reminding me of these two very relevant papers, both of which are cited elsewhere in the manuscript. But, regarding the shape parameters at UK tide gauges as discussed here, are they known to be negative? The recent CFB2018 report (with project team including Jon Tawn and Kevin Horsburgh) diagnosed a mixture of positive and negative shape parameters. I am not aware of a more recent publication specific to the UK which overturns that result. |
| 1.7 | 69 - give a reference. For example chapter 7 of Pugh and Woodworth (2014). As well as the physics of wind stress etc., there is a general point that there is only so much water in the ocean, so one would imagine any extreme level curve to turn down at some point. | Reference: done. Regarding the curve turning down: I agree. A similar argument would apply to any physical process, I think. I contacted Simon Brown (a Met Office climate extremes expert) about this. He said: "*Ultimately all meteorological processes have a physical limit and so should have bounded distributions. However, we don't have perfect samples and we don't fit perfect statistical models. Both, I would argue, can lead to unbounded models seeming to be the best fit to data. As you say there seems to be no consensus on what to do about this nor even much of a discussion about the problem.*

*My go to example is with wind extremes. An observational record will consist of samples of the warm conveyor belt, the cold conveyor belt and if you are lucky a sting jet. Each of these processes have different extreme behaviour but we fit a simple EV distribution. If there is just one sting jet sample, which*" |

| | | *is way above the others, the resultant EV fit will have a positive shape parameter even if all the three sub-processes have negative shape parameters. The fitted EV model does not reflect the underlying mixture of processes and so when it is fitted we get a distribution that looks unphysical.*

*What happens is that generally there is some pragmatic choice that seems to fit the main objective on the analysis. I suppose being a good scientist one would look at the results with a free shape fit and compare with a constrained shape fit and discuss the merits of each. This can be quite subtle – with my wind example it is not clear that forcing the shape parameter to be <=0 will give you a more physical fit if the error of not including the mixture aspect of the sample is not fixed. It could easily be worse.*" |
|---|---|---|
| 1.8 | 73 .. in [shape parameter] (reference needed. HW21 again?) | I have added a "pers. comm." type citation and a footnote.
A colleague (Simon Brown again) pointed me to the scale-shape compensation issue last year. He can no longer remember the details of the paper where he read it. I had thought to maybe show analytically that the partial cross-derivative of the loglikelihood is usually negative, but it proved to be beyond my mathematical scope. So, I have settled for a footnote explaining that it can readily be confirmed numerically, and how to do so. |
| 1.9 | I don't understand why in practice you know there is spatial correlation in the location parameters. That can only be in model runs where the datum at every point is MSL. But if you are using real tide gauge data the location parameters will depend on the datums used at each site. (I hope you see what I mean.) | Yes, I do see what you mean. I have modified the caption of Fig 1 to clarify that we are dealing with the shape parameter of skew surge here: hence not dependent on the datum. |
| 1.10 | 79 - why 'vector'? It seems an odd word to use here. | Now "data" |
| 1.11 | 82 - say 'For the model data at each tide gauge site'. To make it clear you are using just the short model data sets here and not the 484 year set mentioned later. | I am using the 484 year data, There is no short model data set. Have modified the m/s: "The simulation takes atmospheric data from a free-running 484-year climate model control run..." |
| 1.12 | 90 - from any other site. (?) | Actually from any site. Could in theory be the same one very occasionally, but since they are shuffled at random many times this does not materially affect the result. |
| 1.13 | 104 - the long run of 484 years. And this is for the 44 (?) tide | Yes. Have adjusted caption. |

| | gauge sites | |
|---|---|---|
| 1.14 | 112 - .. not Gumbel-distributed as was known previously. | Changed to "re-emphasising…" |
| 1.15 | Figure 2 (a) and (b) should have (m) on each axis | Thank you. |
| 1.16 | line 4 of caption - .. the site of the 44 (?) tide gauges on .. | Yes. Have adjusted caption. |
| 1.17 | section 2.4 - I got the idea of this section although you have to read it a few times. It would help to fully explain things. For example, what does 'standard-uniform' (line 121) mean? | Have added a note explaining "standard uniform", and reworded parts of the section. Have also tried to further emphasise that this is just a short informal description, for ease of reference, of the procedure that VdBK08 used. Have directed the reader to VdBK08 for full details. |
| 1.18 | 126 - … from a given site conforms to a precise GEV distribution. | Reworded. |
| 1.19 | 130 - .. depends on the three GEV parameters. | Reworded. |
| 1.20 | 133-134 - standard-uniform (as above) | Thank you. |
| 1.21 | 151 - an average (?) optimum .. They preferred | They use the word "optimal". I believe they performed their test with a fixed shape parameter across all sites, varying the shape parameter to see which value gave the "best" plot (closest, in some sense, to x=y on their plot). |
| 1.22 | 155 - simulation as represented in Figure 1 (presumably) | Yes. Reworded. |
| 1.23 | Figure 3 - I don't understand why there are 4 plots here. Shouldn't there be 8? You have tide gauge data (shown here) and line 155 says you use model data also, so you need another 4 for the model data? | Figure 3 shows simulation-based results in top two panels and observation-based results (i.e. tide-gauge-based) in bottom two panels. |
| 1.24 | title caption should be VdBK and not VdB&K to be consistent with the text But I would remove that anyway and just have QQ plot to be consistent with PP plot on the right. Preumably the dots are ordered so as to be monotonic. Define in the caption the delta symbol on y-axis for QQ (differences at the outliers). Finally I don't understand why you call them 'theory'. | Have changed to VdBK08 to be consistent with the text. Yes, the dots are ordered so as to be monotonic (i.e. they are ranked). This is the usual approach in this kind of plot, as I understand it. The X-values of the PP plot are the expected values for a ranked set (of size m) of standard-uniform data: $i/(m+1)$ (i: 1 to m). The Y-values are what we have for our sample set, given the fit. The QQ plot in this case is the same data, with both X- and Y-values transformed by $-\log(-\log(.))$ as described in the text. I have used the word "theory" for consistency with VdBK08: it shows the relationship you would see if every element of your sample took its expected value. |
| 1.25 | caption line 1 - this should be reworded as you say above for both that QQ and PP derive from VdBK | They are not both derived from VdBK08. See the line in the text which says "Instead of the plot described above, they…" VdBK08 show only what I call the QQ plot. They do not |

| | | | show the PP plot. |
|---|---|---|---|
| 1.26 | caption line 2 - at the 44 (?) sites of UK ... | | Yes. Done. |
| 1.27 | 172 - in general zero, as known already (refs). | | Added. |
| 1.28 | Figure 4 - I thought you were using 44 sites (see caption figure 5). This should be mentioned at the places in the text I pointed out above. However here in Figure 4 there are 46 locations given. | | Amended: now 44. |
| 1.29 | 188 - why did CFB2018 take +0.0119 as its prior shape parameter when all the evidence from previous publications and your Figure 1 has it negative? And at line 192 why did you use a prior of +0.0119 ? | | Good question! This is discussed by Jon Tawn and Eleanor D'Arcy in section 3.3 ("Penalised Likelihood ") of their comments on HW21, which are in the public domain at

https://nhess.copernicus.org/preprints/nhess-2021-184/nhess-2021-184-RC3-supplement.pdf

The relevant paragraph says:
*"The prior that was selected in the CFB2018 work was not subjective in the traditional sense of a subjective prior in Bayesian methods. It was actually a data-based prior which corresponds to an empirical Bayesian prior, using all the information that separately estimated shape parameters for UK skew surge provide. The effect of this was simply to move shape parameter estimates more towards the UK average, with the larger changes coming for sites with shorter record lengths."*

Considering the scatter plot (Figure 1 of the manuscript under review here), as you say an overall negativity is seen in the shape parameters. But that is only in the shape parameters diagnosed **from the simulation** (Y-axis), **and not** from the tide gauges (X-axis). Even without the penalty function, the unconstrained shape parameter estimates based on the tide gauge data have a **positive** mean (+0.0119). Their estimates are shown in figure E.1 of the CFB 2018 report https://www.gov.uk/government/publications/coastal-flood-boundary-conditions-for-uk-mainland-and-islands-design-sea-levels
Their prior is formed from the distribution of their unconstrained estimates.

I used a prior with a mean of +0.0119 to be consistent with the CFB2018 approach.

Incidentally, the negative bias in the simulation-based |

| | | shape parameters compared to the observation-based remains unexplained. It is discussed at some length in HW21. I have checked that it is not simply a sign issue (unfortunately authors and statistics packages use differing conventions regarding what is meant by a positive shape parameter). |
|---|---|---|
| 1.30 | Could you explain Figure 5 a bit better? If the data really has a non-zero shape, and the choice of prior is reasonable, then wouldn't you expect the right-hand side to be tighter than the left for Gumbels? | I have added the following phrase: "*This figure shows that, even though we believe that the data represent distributions with non-zero shape parameters, the likely inaccuracies associated with unconstrained shape parameters are more serious than the likely inaccuracies associated with the over-constraint of insisting the shape parameters be zero (Gumbel fitting). In other words, we see the importance of choosing an appropriate prior constraint on the shape parameter, for typical real-world record lengths*."
As the existing text already explains, we are choosing to take the constrained fit as "truth" for the purpose of this experiment. |
| 1.31 | 198 - .. is negative in common with most UK sites (Figure 1) and worldwide (Marcos and Woodworth, 2017; Wahl et al., 2017). | Again, I'm not so confident about asserting that here. As mentioned above in response to your comment number 29, the negative shape parameters in my Figure 1 are only for the simulation. See also Fig E.1 of the CFB2018 publication: they diagnose a fairly balanced distribution of +ve and -ve |
| 1.32 | Figure 6 left - the Hinkley plot is described in some detail in Batstone et al. (2013) | Thanks. I have added a citation. Although, incidentally, I regard Batstone et al. as having been largely superseded by CFB2018. |
| 1.33 | Acknowledgements - define BEIS and Defra | Done. |
| 1.34 | 221 - Climate, 231 - Research Letters, 235 - Communications | Amended. Thanks. |

Author responses to Review by John Hunter of "Technical Note: Tail behaviour of the statistical distribution of extreme storm surges". Tom Howard. April 2022. Please see also the proposed appendix.

| Ref. | The paper is generally well written, the problem is described clearly in the Introduction and the results are summarised clearly (although perhaps too briefly for some) in the Conclusions. However, the middle part (the bulk of the manuscript) does contain some quite complex concepts and methods, which could, I feel, be helped with a little more explanation and perhaps a few equations.
The following line-by-line comments are reasonably minor - if they are attended to, I would have no objection to the manuscript being published. | Thank you for your review. I have addressed each of your points below, in most cases modifying the manuscript. |
|---|---|---|
| 2.1 | Page 2, lines 27-29: given that the previous sentence mentions both the annual-maxima and peak-over-threshold methods, this sentence should mention the peak-over-threshold equivalent of the GEV, which is the Generalised Pareto Distribution (GPD). | Done. |

| | | |
|---|---|---|
| 2.2 | Page 2, line 37: I generally dislike unnecessary abbreviations; the reference "Howard and Williams, 2021" (here abbreviated to "HW21") only occurs five times in the manuscript, but I leave this decision to the editor. Likewise, "Van den Brink and Können (2008) (henceforth VdBK08)" on Page 6, line 114. | I am happy with either abbreviated or full form, at editor's discretion. |
| 2.3 | Page 3, line 53: it would probably be good to define the "skew surge" at this stage. | Done. |
| 2.4 | Page 3, line 59: the author uses the terms "geographical variations" and "spatial correlation" throughout Section 2. I don't think this is an appropriate terminology, because there is no actual "geographical" or "spatial" coordinate used in the analysis (for example, there is no analysis of the correlation of the shape parameter with latitude). It just happens that the modelled and observed data come from the same locations - the actual locations are just "labels" which relate modelled data points to their equivalent observational data points. I don't think this, in any way, affects the results but the author may want to clarify his terminology. | Thanks. Phil Woodworth made a similar comment. In response, I have included a text description of the spatial variations and a more clear pointer to the plot in HW21 which shows the variation around the UK coastline. I feel that having done this, it is acceptable to continue to use the phrases "geographical variations" and "spatial correlation". It would, of course, be possible to use phrases like "site-to-site" variation and "inter-site correlation", but I fear that this might "muddy the waters". |
| 2.5 | Page 5, lines 104-112: I got a bit lost here - it would be good if the author showed what he did by using a few equations. It also seems strange that he says that "the RMS difference based on the real data is more than 6 standard deviations away from the mean of this random distribution", when Figure 2 (a) seems to show that most points are within the 95-percent (approximately 2 standard deviation) uncertainty range of the red line. I am not saying the author is wrong - only that he needs to provide enough equations to convince me that he is right. | I have added an appendix detailing the method described in these lines. I will upload a copy of the appendix as part of my author responses (since uploading of the revised manuscript is prohibited).

**TO DO: check appendix is uploaded.** |
| 2.6 | Page 6, line 120: I found the "PIT" at the start of the line very confusing, until I read the whole paragraph and realised that it stood for "probability integral transform". I can't see the point of this initial "PIT" - I'd omit it altogether. | This is one of those things that evolved from a previous version. I can see how it could confuse a reader coming to it for the first time, so I have removed it as suggested. |
| 2.7 | Page 6, lines 128 and 134: it is a bit confusing that the author uses $U_i$ to represent the PIT-transformed $Y_i$ and $U_j$ to represent the PIT-transformed $M_j$ - it would be clearer if a letter other than "U" was used in the second instance. | I would prefer to stick with using U for both, as this underscores the fact that both $U_i$ and $U_j$ are expected to be uniformly distributed. This gives a consistency throughout the article: Y is a distribution of annual maxima from the shelf sea model (assumed GEV), G is Gumbel-distributed, U is uniformly-distributed, and R is an unspecified known distribution. |
| 2.8 | Page 6, line 129: it would be good to expand on "standard-uniform" (e.g. $f(x) = 1$ for $0 <= x <= 1$; $f(x) = 0$ for $x < 0$ or $x > 1$). | Thanks. Have done something like this in response to a previous review comment from Phil Woodworth. |
| 2.9 | Page 8, Figure 3: I may be being pedantic here, but I feel that it would be easier to understand if the PP plots (which were introduced first, on line 142) were on the left and the QQ plots (which were introduced second, on line 144) on the right. | Agreed; have swapped them round. Pedantic, but in a good way. |
| 2.10 | Page 8, Figure 3: the labelling of the vertical axis of the left-hand panels is obscure - it is presumably the gumbel variate of the samples, and so should be labelled something like "Gumbel variate (samples)". For consistency with the right-hand panels, the horizontal axis should be labelled something like "Gumbel variate (theory)". It would also clarify the panels if the labelling of the tics was consistent for both vertical and horizontal axes. | I used the "Delta $X_n$ hat" notation for consistency with Van den Brink and Konnen 2008. I have added a more friendly label now, and made the ticks consistent, as suggested. |
| 2.11 | Page 10, line 185: this is the first occurrence of "CFB2018", which is not defined anywhere. It is presumably "Environment Agency: Coastal Flood | Thank you for pointing this out. I have now defined CFB2018 at first occurrence (earlier in the paper) as it occurs many |

| | Boundary Conditions for the UK: update 2018" - this should be defined here. | times. |
|---|---|---|
| 2.12 | Page 12, lines 203-207: I love the brevity and clarity of the Conclusions, although some may find it too terse. | I hope readers enjoy the brevity. |

Author Response to Anonymous Review submitted 21 April 2022

| Ref. | Anonymous Reviewer | |
|---|---|---|
| | Thank-you for this technical note, which I believe will be helpful in analysing return periods of UK gauges and is worth publishing in OS. A little work is required to make it easier to understand, especially for readers considering how it applies to sites outside of the UK network. Some suggestions: | Thank you. |
| 3.1 | The paper could be made easier to follow independently. It currently relies on too much cross-referencing to other papers particularly H&W21. | I would prefer to keep this tech note short, at the expense of some cross-referencing. HW21 is open-access and CFB2018 is in the public domain. Please see also 3.2 below. |
| 3.2 | Please provide equations for GEV/Gumbel, and GPD that is mentioned later. | Please see my Ref 2.5 above: appendix now added.
I did also include the GEV/GPD equations in one version of the paper, but I didn't like it. The problem is that I want to refer specifically to the effect of the shape parameter on the return level curve. The usual expression of the GEV distribution only describes the return level curve in an indirect way (it tells us about the water level and the probability of finding an annual maximum less than that level, as opposed to the level and (the log of) the average recurrence interval of that level). This is addressed at some length in HW21, and is also covered in sources such as textbooks and Wikipedia. I would prefer for this article to remain a short technical note, rather than include a lot of recapitulation of established theory. I think this is a decision for the editor. |
| 3.3 | Fig 1 only makes sense to someone very familiar with the names and locations of the UK TG network, and even then it is hard to determine whether there is a spatial relationship or whether the relationship is due for example to tidal range. Perhaps using the colouring to group neighbouring gauges in clusters would be useful? And certainly a map. | Please see my response to 1.2 and 2.4, above. |
| 3.4 | But I would also like to see some evidence of the spatial correlation of mu and lambda, perhaps some maps indicating all three parameters, as fitted to the data as it stands, and then under the experimental conditions? Or plotted with the coastal position as an axis - I see from panel 1c that you have already ordered the sites clockwise around the coast. | These plots appear in HW21.
I would rather not reproduce large amounts of information which is freely available elsewhere (HW21 is open-access). |
| 3.5 | line 73: artefact since we're in British spelling | Done. |
| 3.6 | line 113: You show that the fitted scale parameter | Please see my response to 2.5, |

| | | |
|---|---|---|
| | lambda, assuming a Gumbel distribution, is slightly higher than a Gumbel should allow? What does this imply, physically? How does assuming a Gumbel therefore bias the extrapolation? Perhaps work through an example? ... ah this comes in figure 6, thanks. It might be easier to understand the general argument if you brought fig 6 forward. | above, and the new appendix. The "slightly higher" effect that you note is coming from the fact that the simulated annual maxima have predominantly negative shape parameters. Fitting a distribution with a negative shape with a Gumbel would in general give a positive bias to the extrapolation, I think. |
| 3.7 | Figure 3: I'm afraid I don't really follow what is going on here, this plot is not well explained. | Now revised. Please see 1.17 thru 1.25 above, and 2.9 and 2.10 |
| 3.8 | Fig 6: It is quite concerning that Hinkley shows such a large uncertainty at very long return periods depending on the method, considering the reason for the gauge! If this is very atypical, a more typical example would be illustrative. Probably not Bournemouth, which has its own unusual challenges. | The departures for all sites are already shown in figure 5. |
| 3.9 | line 202: Can you give any guidance on what constraints should be applied to the GEV shape parameter in practice? Is it the same at any site? If not, what varies? | This is beyond the scope of this tech note, although I hope that this note and HW21 might help to inform the next generation of CFB. I regard CFB2018 as the current best practice. The prior used by CFB2018 is described in the tech note. |
| 3.10 | Conclusions: I agree with the other reviewer that it is refreshing to see such succinct conclusions! | Thanks. |
| 3.11 | References: There are several missing DOIs. | Thanks: ten DOIs inserted :) |
| 3.12 | Data: data should be open and links provided. | As I understand it, this is not a requirement for Ocean Science. Curating all of the underlying data would be a significant overhead, but the points shown in the plots could be included in an appendix without too much trouble, if that is helpful? ...Editor's decision please. |

---

## Author Comment (AC4)

**Appendix A: Statistical test associated with figure 2**

Let m be the number of sites considered.

[revised manuscript text omitted]

---

## Author Comment (AC5)

**Appendix A: Statistical test associated with figure 2**

Let $m$ be the number of sites considered.

[revised manuscript text omitted]

**14**